# Motivations and Barriers for Sheep and Goat Meat Consumption in Europe: A Means–End Chain Study

**DOI:** 10.3390/ani10061105

**Published:** 2020-06-26

**Authors:** Serena Mandolesi, Simona Naspetti, Georgios Arsenos, Emmanuelle Caramelle-Holtz, Terhi Latvala, Daniel Martin-Collado, Stefano Orsini, Emel Ozturk, Raffaele Zanoli

**Affiliations:** 1Department of Materials, Environmental Sciences and Urban Planning (SIMAU), Università Politecnica delle Marche, Via Brecce Bianche, 60131 Ancona, Italy; mandolesi@agrecon.univpm.it; 2Faculty of Veterinary Medicine, Aristotle University of Thessaloniki, P.O. Box 393, GR-54124 Thessaloniki, Greece; arsenosg@vet.auth.gr; 3The French Livestock Institute/Institut de l’Elevage (IDELE), Campus INRA—Chemin de Borde Rouge, CEDEX, BP 42118-31321 Castanet Tolosan, France; emmanuelle.caramelle-holtz@idele.fr; 4Economic Research, Natural Resources Institute Finland (Luke), Koetilantie 5, 00790 Helsinki, Finland; terhi.latvala@luke.fi; 5Animal Production and Health Unit, Agrifood Research and Technology Centre of Aragon (CITA), Gobierno de Aragón, Avenida Montañana 93, 050059 Zaragoza, Spain; dmartin@cita-aragon.es; 6Organic Research Centre, Trent Lodge, Stroud Road, Cirencester, Gloucestershire GL7 6JN, UK; stefano.o@organicresearchcentre.com; 7Department of Agricultural, Food and Environemntal Sciences (D3A), Università Politecnica delle Marche, Via Brecce Bianche, 60131 Ancona, Italy; ozturk@agrecon.univpm.it

**Keywords:** laddering, consumer attitudes, meat, food consumption, small ruminants, consumer perceptions

## Abstract

**Simple Summary:**

In Europe, human consumption of sheep/goat meat is lower than for other types of meat (e.g., chicken, beef). This study contributes to a better understanding of why/why not sheep/goat meat is consumed in Europe, and which are the relevant attributes, situations associated with small ruminants’ meat consumption by consumers.

**Abstract:**

This international study involving seven European countries (Italy, UK, Finland, France, Spain, Greece, Turkey) was conducted to explore the relevant reasons that affect both consumer and non-consumer perceptions of sheep and goat meat. Laddering and means–end chain theory were applied. The results indicate that consumers associate sheep and goat meat with a unique taste, authenticity and natural production, linked with values such as health and enjoyment of life. In contrast, non-consumers mainly feel disgusted when they think and feel about these meats, and do not associate any specific health benefits to their consumption, disliking their taste, odour and fat content. This study is based on qualitative research. Only analytical generalizations are possible, expanding the theory on what drives consumer behaviour when purchasing meat. No previous means–end chain studies are available in the literature regarding small ruminants’ meat consumer preferences.

## 1. Introduction

Meat is an essential source of protein in the human diet, and as such it has a key role in our food culture. Global meat production has been undergoing a moderate increase (by 1.25% in 2017; up to 323 million tons). This has been driven in particular by higher consumption in developing countries, where beef and poultry meat are the largest meat markets [1], while sheep and goat meat remain a niche market in the consumer daily diet. Globally, in 2018, the small ruminants sector included around 1209 million head of sheep and 1046 million head of goats, representing a little less than 12% of total Livestock Units (LU) [2].

In Europe, in 2018, the sheep and goat populations included approximately 146 million head, representing approximately 11% of total LU [2]. For the sheep sector the most relevant product in economic terms is sheep (lamb) meat, followed by dairy products and wool. In the goat sector, milk and cheese are the main products on the market [3]. Despite the low relevance in terms in the share of LU, the small ruminant sector helps to maintain local traditions and to preserve the environment in many marginal rural areas. Nevertheless, the European sheep and goat meat sector increasingly suffers from international competition and weakness in the internal market. On the supply side, the seasonality of consumption and the carcase imbalance caused by only few cuts being demanded on the market (i.e., leg, chops and, to a minor extend, shank) create market opportunities for a large share of imported products. Inadequate production levels translate into the highest world importation of sheep and goat meat by European countries, which come mainly from New Zealand and Australia [1]. On the demand side, the lack of appeal of sheep and goat meat, especially by younger consumers, and the switching of meat consumption from red meats to pork and poultry meat are all factors contributing to decreasing the consumption of sheep and goat meat in Europe.

On the basis of these issues, this study was designed to provide deeper understanding of consumer perceptions of sheep and goat meat, as well as the motivations and barriers experienced by consumers when approaching these products. This study reports the empirical results of a combination of two qualitative approaches that were carried out simultaneously in seven European countries: focus groups and means–end chain (MEC).

A literature review of consumer studies in this field shows that little information is available, especially at a pan-European level. Peer-reviewed studies were retrieved from the SCOPUS database in November 2019 using a hierarchical search procedure for titles, abstracts and keywords: (1) “consumer” AND (2) “meat” AND (3) “sheep” OR “lamb” OR “goat” OR “ewe”, AND (4) “preference” OR “perception”. This search yielded 181 documents of potential relevance, of which 172 were in English (with a few in Portuguese, Spanish, German, Turkish). In most cases, the final selection required reading of the full text to decide on the selection or exclusion of the papers. Finally, only 20 papers were relevant to our research purposes here, which referred to the general consumer [4,5,6,7,8,9,10,11,12,13,14,15,16,17,18,19,20,21,22,23]. 

It is relevant to discuss briefly the studies that were found regarding consumer awareness, attitudes, beliefs and preferences, and the overall consumer acceptance of sheep and goat meat. When looking at consumer motivations, despite the extensive literature associated with meat consumption [24,25,26,27,28,29,30,31,32], only a small range of studies have considered sheep (mainly lamb) and goat meat [5,11,17,20,22,33,34]

According to these studies, consumers distinguish lamb meat from other kinds of meat for specific reasons, such as a higher price, an intense or unique taste, limited possibilities in terms of processing and transformation, and the time needed for preparation, with a lack of versatility in terms of cooking [22]. A combination of different variables can also affect the consumer sensory experience [35]. Overall, the typical flavour of lamb meat was considered one of the intrinsic key characteristics of consumer perception [20,36], sometimes along with its textural attributes [9]. The hedonic evaluation of lamb meat by consumers in European countries resulted in segments with different taste preferences: consumers preferring a “milk or concentrate” taste, and others preferring a “grass” taste [9,23]. Angood et al. [8] provided evidence of better hedonic perception (i.e., actual liking) of organic lamb chops versus conventional lamb chops. Additionally, when goat meat was considered, consumers perceived the typical flavour of this meat as different from that of beef, even when goat meat was served with seasoning to mask its taste [13].

According to Spanish consumers, extrinsic quality cues are the most relevant attributes when selecting lamb meat. In the first place, there is the type of lamb meat (e.g., suckling, Ternasco, Feeder lamb), which is a consequence of the type of breeding and the age of the animals at slaughter. The second most valued characteristic was the origin of the meat for regular consumers, and its certification for occasional consumers. Due to the sense of identity held by many consumers, the meat of their own country was reported to be the most preferred characteristic, as also for other European countries [6,10]. 

A few years later, Bernués et al. [22] analysed Spanish lamb meat consumers again, according to their convenience orientation, and they identified different consumer profiles. Overall, the presence of quality labels, local origin and feeding systems (with a preference for pasture or green forage) were highly appreciated by the majority of the consumer profiles identified. More recently, de Andrade et al. [5] showed that there is strong association between lamb meat and particular special eating occasions, which might also explain why this meat is not consumed on a daily basis.

Given these previous research findings and the wide dynamics of how consumers think about sheep and goat meat, the present study combined two qualitative methodologies: focus groups and a MEC approach. Focus group discussions [37] were aimed at an exploration of the consumer attitudes and preferences towards sheep and goat meat products and their shopping habits, and to identify consumer needs, as well as their attitudes towards novel recipes and innovations from their point of view. Consumer motivations and barriers for buying sheep and goat meat were further investigated via the MEC approach [38].

To the best of our knowledge, only one study has analysed consumer motivations with reference to lamb meat [12]. In this study, elderly Australians who consumed red meat were asked to give the reasons for their choice for different cuts of lamb or beef, although the analysis do not specifically tackle relevant consumer reasons for eating lamb meat. As for other studies on meat in general, where it has been shown that healthiness and sensory attributes motivate red meat consumption [39], Flight et al. [12] did not investigate the barriers or negative associations that consumers might have in relation to sheep and goat meat consumption.

The rest of the present paper is organized as follows. In Section 2, the methods applied are described. Section 3 presents the results, while in Section 4 the results are discussed, which is followed by the conclusions.

## 2. Materials and Methods 

All subjects gave their informed consent for inclusion before they participated in the study. The study was conducted in accordance with the Declaration of Helsinki, and the protocol was approved by the Executive Committee of the project ‘Innovation for sustainable sheep and goat production in Europe’ (iSAGE) (grant agreement 679302) in accordance with the “Ethics for specific socio-economic activities” guidelines reported in the Deliverable 7.7 to the Research Executive Agency.

A total of fifteen focus groups were conducted in each of the seven selected countries, namely Spain, Finland, France, Greece, Italy, Turkey and the UK, between November 2016 and February 2017. A total of 140 participants were recruited from outside supermarkets or other shopping outlets, using existing datasets and through snowballing. All of the selected participants were responsible for their food shopping and were consumers of both sheep and goat meat products. For each focus group, both women and men were recruited (male/female, 30% to 60%). All of the participants were between 25 and 65 years old. At least 30% to 60% of occasional/regular meat consumers were included. Table 1 shows their key characteristics by country. Consumer classification was based on self-reported frequency of consumption: regular consumers were those who reported consuming sheep/goat meat at least one time per month; occasional consumers were those who consumed less than this, but not never or very seldom. 

Focus group research, as with case studies and other qualitative research methods, is conducted to gain a more complete understanding of a particular topic [40], such as consumer motivation, barriers and perceptions on sheep and goat meat in Europe. Laddering refers to “an in-depth, one-on-one interviewing technique used to develop an understanding of how consumers translate the attributes of products into meaningful associations with respect to self, following Means–End Theory” [41]. The goal of both methods is not to achieve standard statistical generalizations but to go “in-depth” and achieve conceptual or analytic generalization; that is, (a) corroborating, modifying, rejecting or otherwise advancing theoretical concepts, or (b) providing new concepts, ideas, theories, and hypotheses [42]. This form of generalization is driven by semantics rather than statistics [43].

The focus group discussions in each country were conducted following common and agreed guidelines. A training workshop and some pre-test sessions were carried out to test the discussion guides in the different countries. The guide for the moderator included an introduction section to define the purpose of the study, and probing questions, expected outcomes, and a list of innovations in sheep and goat meat production/ processing for discussion. To avoid bias in the common flow of discussion, the participants completed the laddering task (see below) shortly after a brief introduction and an icebreaker, to let the consumers introduce themselves. Then, following the guidelines, the discussion started to investigate the consumer attitudes and beliefs, with an examination of their relevant likes and dislikes towards sheep and goat meat. Finally, new strategies and innovations to increase the consumption of these kinds of meat were discussed. The guide also included the recruitment criteria and instructions for the transcription and data analysis, plus instructions for the laddering task. The moderators directed the flow of the discussion for each focus group, accompanied by an assistant who took notes and ensured that all of the important issues were discussed. The focus group conversations were audio recorded, and in some cases also video recorded. In most countries, a small incentive (20–40 €) was offered to the participants to compensate for their time. In Turkey, a quick breakfast or lunch was often offered to the participants to encourage them to attend. Both the focus group discussions and the laddering data were collected in the language of each country, and the results were translated into English.

The laddering task embedded in the focus group discussions is based on the Means–End Chain (MEC) theory [38]. This theory and its interviewing technique (i.e., laddering) have been widely tested [41,44,45] and support improved understanding of why specific attributes are relevant for consumer choices, or why despite positive attitudes towards sheep and goat meat, some consumers are not purchasers of these products. 

Given that the laddering approach is well known and widely used in food studies [26,46,47,48,49,50,51,52,53,54,55], its presentation here is not exhaustive, but provides a broad overview of the methodology and introduces the reader to its specific terminology. Readers who are not familiar with this approach and are interested in obtaining a deeper understanding are invited to read about MEC theory and the laddering technique in Gutman [38] and Reynolds and Gutman [41]. This theory assumes that consumers choose a product according to the attributes that provide them with specific valuable benefits, as those that will satisfy their values (as a means to an end). The model makes it possible to uncover the latent motivations of consumers when purchasing goods.

The MEC theoretical approach to consumer motivations uses the laddering technique [41,44] to reveal how consumers associate attributes, consequences (benefits) and values, and to identify their MEC cognitive structure. The result is a knowledge structure that links motivations, i.e., consumer knowledge of product attributes, to consumer self-knowledge (through personal consequences and values) [38,56]. According to the laddering technique, three steps are necessary to build a relevant MEC [41]. First, consumers are asked to provide product attributes and to sort these based on their viewpoint (i.e., elicitation). Secondly, the personal consequences and values are gathered using the iterative question, “Why is this important to you?”, to build any individual means–end ladder. Answer after answer, consumers are naturally guided to build their own sequences of their attribute–consequence–value chain. Thirdly, the consumer replies are coded and analysed, and an implication matrix is created from all of the data collected. The implication matrix is a square matrix that reports the frequency of the connections between single categories of attributes, consequences and values, and it is used to build the hierarchical value map (HVM) that shows associations between attributes, consequences and values.

In this survey, a hard-laddering approach was preferred to the soft-laddering approach. This was defined in relation to the cross-cultural nature of the study and the difficulty in finding well-trained interviewers in each country [57], and also to the hypothesis of low involvement of the consumer in meat choices [58]. The timing restrictions and the knowledge that a written questionnaire increases the objectivity of the results also supported this decision. 

Two laddering questionnaires were developed: one to investigate the consumer motivations to consume sheep and goat meat, and one to investigate the non-consumer barriers to sheep and goat meat consumption. Both of these included three sections: the Instructions; the Eliciting section for extraction of desirable (or undesirable) attributes of sheep and goat meat; and the Iterative laddering questions to understand the motivations (or barriers) for the three main attributes. In each country, the barriers questionnaire was completed by 12 non-consumers of lamb and goat meat using similar recruitment quotas to those of the sheep and goat meat consumers reported above. Among the characteristics of the non-consumers, who only completed the laddering task, their age was a little lower compared to the consumers. The group of non-consumers was composed of: females, 57.5%; 25–45 years old, 67.8%; and workers, 69.0%.

All of the laddering questionnaires were translated for the collection of the data in the language of each country, and then each relevant ladder was translated back to English and entered into the software for analysis (Mecanalyst cloud 2.0, SKYMAX_DG, Italy, 2016). The final coding of the ladders was centralized and was performed by two independent assessors to determine and define the inter-coder reliability [59]. Inter-coder reliability largely exceeded the recommended guidelines.

Finally, 226 completed laddering questionnaires were collected and contained valid responses: 139 laddering questionnaires that explored the motivations related to sheep and goat meat, and the 87 laddering questionnaires that explored the barriers. To choose the most informative and easy-to-interpret solution, different cut-off levels were applied (i.e., 6 for motivations; 4 for barriers). Every effort was made to keep at least 40% of the total links in each map. The relevant maps provide graphical illustrations of the networks of cognitive links between the sheep and goat meat characteristics and the consumer (at different levels of abstraction: attribute–consequence–value). These were analysed at an aggregate level and as country differences (as for the related hierarchical value maps) and used as the basis for the in-depth consumer analysis. 

The abstractness and centrality indices were also used to examine the role of each variable (attribute–consequence–value) in the cognitive structures. According to previous studies [44,60], the abstractness index ranged from 0 to 1 and provided a measure of the percentage of times the variables served as means (or ends) in the HVMs. The higher the abstractness score, the more the variable represented an end of the variable connections over other variables. The centrality index comes from network theory [61], and shows the percentage of connections in the structure that run through that particular variable. The higher the centrality index, the larger the number of links with other relationships. 

## 3. Results

The focus group results were analysed at the country level to reduce the time and translation costs, and to avoid overlooking aspects that required deep understanding of the language and culture. Information related to body language was not included, although relevant statements were captured and used to better represent the participant viewpoints.

A thematic analysis was performed, and the aggregated results are presented here. Detailed analysis of the transcripts yielded the following topics: consumer preferences and beliefs, attitudes (i.e., likes, dislikes), shopping habits, and suggested marketing strategy to increase consumption. A specific comment on the participant’s acceptance of possible innovations in the supply chain ended the discussions.

The results of the laddering task were analysed at a meta-HVM level. All of the country-level data were merged into unique data files to reveal the cognitive structures concerning the relevant concepts for the motivation or barriers to consumption of sheep and goat meat. These meta-HVMs combined and integrated, on common grounds, all of the original HVMs obtained at each country level. This paper reports and discusses the results at an aggregate meta-level. Each level of abstraction in the HVM (i.e., attribute, consequence or value) is identified by the colour of the node. For each map (Figure 1 and Figure 2), the values are shown at the top in purple (VI) or violet (VT), the consequences (both CF and CP) in the middle in green, and the attributes at the bottom in yellow (AA) or orange (CA). In the HVMs, the thickness of the lines represents the frequency of the associations between the attributes, consequences and values. Table 2 provides a description of the HVM content collected for both motivation and barriers to consumption.

In the following sections, the main findings of both the focus group discussions and the laddering questionnaires are reported. First, the participants’ preferences, attitudes and beliefs towards the consumption of sheep and goat meat are presented. Secondly, the participants’ shopping habits, opinions on the presented innovation strategies and suggested marketing strategies to increase consumption are reported. The last paragraphs illustrate the motivations and barriers as represented in the meta-HVMs. 

### 3.1. Participant Preferences, Attitudes and Beliefs towards Sheep and Goat Meat

It must be acknowledged that as the majority of the participants declared, they had never tried goat meat and only a few were interested in tasting it. Therefore, most of the focus group discussions were centred on lamb and sheep meat. Most participants usually consumed lamb meat more than sheep or goat meat. Therefore, concepts and viewpoints expressed mainly related to lamb meat. The fact that goat meat was seldom specifically mentioned is—per se—a result. It shows how little place this meat type has in the mind of consumers.

In general, small cuts (e.g., chops, ribs, shish kebab, köfte, arrosticini) were preferred, because they were considered easier to prepare and cook, especially among the younger participants: “I prefer arrosticini when I’m with my friends, but for my family and me, I usually cook the lamb in the oven”. Nevertheless, although a clear preference for small cuts was reported, one participant warned about single-portion packaging as a negative quality cue for this kind of meat: “If small quantities and pre-packed meat means that this meat is imported, then I prefer to maintain the same consumption pattern as today”.

For the consumers, the “less industrialized” and the “free-range” production systems that are traditionally associated with the sheep and goat meat sector make these meats “more natural” and “genuine”, as compared to cattle and poultry. As one participant stated, “I never buy chicken meat; I prefer lamb or sheep meat for their genuineness and taste”. Lamb meat was also often perceived as “tasty”, as having a “good smell”, and as “tender”. Many believed that these intrinsic characteristics of the meat would depend on the type of livestock production system used.

The majority of participants had a free-range ‘perception’ for this kind of breeding. This aspect was highly appreciated, especially because participants believed that it had a positive effect on animal welfare and meat quality. In general, the consumers perceived any aspects related to both animal welfare and feeding as indicators of the healthiness of the animals (and hence the meat). The health aspect also emerged among consumers in relation to the fat content. Some participants believed that sheep and goat meat had a lower fat content compared to other meats, and was particularly suitable for children or people with diseases. Some other participants also believed that “Lamb production has less [environmental] impact than other meat production”.

A preference for local meat was noted (except in Finland, where this kind of meat is only imported). Participants often mentioned the attribute “local”, and referred to the origin of the meat as an incentive for purchasing lamb or goat meat, and also as an indicator of freshness and quality. 

On the negative side, many participants, and even those from countries like Italy, UK, Spain and Greece, where there is a long tradition of lamb and sheep meat consumption, believed that these meats were difficult to cook and required high cooking skills. Some mention was made of the efforts of cooking a whole lamb and the long time required for preparation compared to other meats. Others believed that lamb and sheep meat took longer to cook: “I prefer sheep leg since it is cooked in a short time”.

Consumers often mentioned price as an important quality indicator. However, all of the consumers perceived lamb and sheep meat as too expensive, and for this reason as not suitable for daily consumption. The possibility of reducing prices was suggested as a strategy to increase lamb meat consumption, but the majority of consumers did not positively accept this solution.

### 3.2. Participant Shopping Habits towards Sheep and Goat Meat

In all of the countries, the lamb consumption was mainly occasional and seasonal—it increases during specific times of the year (e.g., Christmas, Easter, with the exception of Turkey)—and for special occasions—“Lamb is the perfect Sunday meal” or “Good for special occasions like Christmas and Easter”. Only respondents from Mediterranean countries (Italy, Greece, Turkey) reported cooking lamb meat more often during the week at home as a substitute for other kinds of meat.

In general, farmers and butchers were the preferred point of purchase, given their perceived higher-quality “halo”. For example, as one participant affirmed, “Butchers owning small shops are the people we know well, and we trust their products”. These traditional points of purchase were also mentioned for the purchase of whole or half lambs and sheep. The participants who preferred to make their purchases in supermarkets mentioned three main reasons: lower prices compared to the local butcher and farmers; perceived high-quality control (for hygiene reasons); and convenience, in terms of less time needed for shopping. Despite this, some consumers declared that they preferred to purchase their meat directly from the farmers or butchers, because they were aware that the meat is of higher quality,

The poor availability throughout the year and the limited variety of some cuts compared to other kinds of meat were also mentioned as important barriers to their consumption.

### 3.3. Acceptance of Selected Innovation Technologies and Innovativeness

The participants highly valued innovations related to quality improvement of pastures; e.g., the use of new plant varieties/species/mixtures (e.g., legumes, high sugar grasses, shrubs), the provision of detailed information (e.g, on food labels), the development of alternatives to antibiotics use (e.g., using natural antimicrobial compounds), and the development of new strategies aimed at improving animal welfare (e.g., use of sensor ear tags as welfare indicators, use of stress-free slaughter).

Among the most negatively perceived innovations were those related to the introduction of new vaccines and practices related to breeding or intervention in the reproduction and slaughtering phases (e.g., developing new breed traits to increase longevity, fertility and health in flocks, or to improve the quality of meat to make it more uniform, lean and tender). However, the participants also rejected those innovations that impacted directly on consumers to increase their consumption (e.g., new recipe books, involvement of chefs to increase product use, home deliveries direct from farms).

### 3.4. Marketing Strategy for Increasing Consumption

The participants would positively accept any strategy that would improve animal feeding and pasture quality (e.g., using new plant varieties, species and mixtures). Many participants affirmed that a possible strategy for increasing the consumption of this meat would be to communicate the benefits related to its consumption, and to provide more detail on food labels. More specifically, consumers were interested in having information about the origin of the meat, and the production methods, traceability and possible positive effects on human health. Some consumers stated that they spent significant time in reading product labels and would like to receive clear and detailed information that would reinforce their positive beliefs of “naturalness” and “healthiness” associated with sheep and goat meat. Additionally, increasing the visibility of this meat (e.g., TV cooking programmes, magazine articles) and communicating the ease of its cooking were considered possible strategies for increasing the appeal of sheep and goat meat products and for improving their consumption.

### 3.5. Motivation behind the Consumption of Sheep and Goat Meat

According to the centrality indices for sheep and goat meat consumption, the most central motivation was the functional consequences of “Tastes good” (0.16), followed by “Eating healthy” (0.12), and the abstract attribute of “High quality” (0.12). Among the attributes of “High quality”, the most important attribute was “Less fat” (0.8) and “Unique taste” (0.7), together with “Freshness” (0.7), followed by “Own health” (0.5), “Food as enjoyment” and “Taking care of family” as the predominant values.

Examination of the final meta-map (Figure 1) according to the longer chains and the strongest connections [62] uncovers the relevant motivations, where two main themes emerge. The first was that personal health benefits come from high-quality and locally produced animals. The second was enjoyment of the food experience by purchasing tasty products with a unique taste. Apart from these, the large number of different attributes and entwined connections show that there are very different consumer viewpoints of their motivations for the consumption of sheep and goat meat.

The meta-map for the sheep and goat meat consumers suggests that the aspects linked to “Food as enjoyment” and “Own health” were the most important motivations for the purchase of these meats. The most mentioned abstract attributes seen to explain the consumer perception of the meat were “High quality” followed by “Unique taste”, “Less fat” and “Tenderness”.

For consumers, meat of high quality means both a good taste experience and eating healthy. Specifically, “Tenderness” and “Unique taste” defined the most relevant hedonistic chain, which leads to “Feel pleasure” and “Food as enjoyment”. The perceived superior taste of sheep and goat meat was described by respondents in terms of “savoury taste”, “tasty”, “intense taste”, “typical flavour”, “stronger flavour”, and “smell and taste typical of the animal it comes from”. In terms of “tenderness”, lamb meat was often perceived as “soft” and “easy to chew and swallow”. More in general, for consumers, both sheep and goat meat satisfied the desire for eating tasty food and had a psychological benefit because it is associated with “eating with pleasure” and “enjoying”, and with “taking care of yourself through your food”. For this reason, consumers also preferred this kind of meat (i.e., as tender, versatile) when they wanted to satisfy guests and family members. Both the good taste and the ease of preparation and cooking promoted the purchase of this kind of meat, as the rest of the family and friends appreciated it (“Others/family like it”).

The quality of the sheep and goat meat was influenced by the “Origin” of the product, which was specified by the concrete attribute “National origin” and by “Animal feeding”. The “Origin” of the meat represented an important quality indicator for consumers. Consumers retained that meat that was safe should be “domestic”, “near my home” or “locally produced”, which implies higher quality controls. “Animal feeding” was also important because the majority of respondents specified that the quality of meat “reflects how the animal was fed”. Another relevant quality cue was meat freshness. The attributes of “Freshness” and “Nice appearance” of the meat contributed to the definition of the idea of a high-quality product; these are both considered as “safety” indicators. Specifically, the “Nice appearance” attribute was sought as a cue to avoid eating (unhealthy) frozen meat. Comments like “deep pink colour prior to cooking”, “the colour not too dark” or “the dark red” gave people the feeling that the meat was “recently slaughtered, and is fresh” and was of better quality. As a consequence, the consumers perceive that the meat is healthy. The quality cue was also linked to “Have information about production/ origin”, which leads to trust in the quality of the product. Consumers expressed their preferences towards a “safer” product, affirming: “We prefer the meat of quality, advised by our butcher”, “We trust the butcher”, or “I trust the relationship with the reseller”.

Eating healthy was important for their own health and also for taking care of others and the family: “Good for the health of my children”, and “I want to teach the meaning of quality of the food to my kids”. For consumers, taking care of others and the family was also linked to the satisfaction of another important value, the “Social harmony”: “If we are healthy, we can be useful to our families and to our country”.

Price was part of a very short chain that went from the “Less expensive” attribute to “Save money”, which shows that this is not a relevant motivation to purchase sheep and goat meat.

### 3.6. Barriers to the Consumption of Sheep and Goat Meat

In terms of the centrality index, three consequences were predominant: “Stay not healthy” (0.18), “Tastes bad” (0.14) and “Feel unsatisfied” (0.11). “Strong taste” (0.1) was the most central among the attributes, and “Own health” among the values.

On an aggregate level, the meta-map of the barriers (Figure 2) to the purchase of sheep and goat meat look simpler, and the chains that contain the strongest connections are shorter than those for the motivations. The map shows that “Own health”, “Taking care of others/family” and “Food as enjoyment” (at the value level) and a feeling of disgust or dissatisfaction (at the consequence level) were the most important motivational drivers that limited the consumption of sheep and goat meat products.

Hedonic characteristics were the main weaknesses in the choice of lamb meat. The main chain started with the “Strong taste” of sheep and goat meat, passed through the aggregating functional consequence of “Tastes bad” (the code with one of the highest centrality indices and the highest frequency of answers, as 53% of respondents), then onto the dissatisfaction (psycho-social consequence: “Feel unsatisfied”) that prevents “Food as enjoyment”. Respondents complained that they felt disgusted or were not satisfied. When they described the meat, they often used words like “gamey flavour”, “too strong taste and smell”, “leaves a bad odour around the house”, or simply “bad smell”. Taste and smell both have a strong influence in the selection of foods. As a consequence, the non-consumers did not purchase this meat, because it was not part of their family habits: “Children don’t like it” or “My friends don’t like it”.

Contrary to consumers, non-consumers considered both sheep and goat meat to be of lower quality compared to other kinds of meat. The “Fatty” attribute was also highly mentioned by non-consumers, and it represented the most mentioned (40%) among the negative attributes for this meat. These respondents believed that sheep and goat meat is fatty, and this had a strong association with a lack of healthiness (“Not eating healthy”, “Stay not healthy” consequences). This attribute was well explained by the non-consumers comments: “It has more fat than other meat”, “Has a high fat content”, “There is usually too much fat” and “It is heavy”. Eating and staying healthy were also related to family approval: “My family complains about fat meat”. 

A minor chain mentioned the low availability (“Not easy to find”) of the sheep and goat meat among the barriers: “Not easily available in the supermarket”, “This meat takes the second place” or “We cannot buy an invisible product”. Respondents also complained about the difficulties they faced when preparing and cooking this meat. The complexity of the preparation was an important obstacle to the consumption: “Very few recipes”, “It’s complicated to cook”, “I do not know recipes to cook it”. The “Not easy to prepare and cook” leads to wasted time: “Cooking lamb is time-consuming—it does not fit into my lifestyle”. At the consequence level, “Not easy to prepare and cook” contributed to an increase in the feeling of no pleasure or not satisfied.

Although the last two chains do not lead to any value, they contain other minor but significant themes: saving money and animal welfare. For non-consumers, the price of sheep and goat meat was perceived as too high (“Too expensive”): “Lamb meat is a product for the rich people”, “High price reduces my food budget”, “I have to spend more money to eat a lot” and “There are other protein sources that cost less”. This chain was relevant, but not long nor central.

Even though the analysis of the results revealed that animal welfare was not a primary barrier, most of the non-consumers added that they feel guilty towards “lovely lambs” (as mainly associated with slaughtering of lambs): “I feel sorry for the animal” and “Feel sick with the idea of eating a young lamb that was killed for that”. More generally, they had a negative feeling towards activities that did not respect the nature of the animals and their living conditions.

## 4. Discussion

The main purpose of this study was to investigate consumer preferences and attitudes towards sheep and goat meat products. This study has identified the most relevant consumer motivations, as well as the barriers to consumption, along with mapping of the linkages from the product characteristics to the benefits, that lead to important consumer personal values. The focus group discussions and the laddering task together helped to gain deeper understanding of the strengths and weaknesses related to sheep and goat meat consumption.

Overall, the HVM for the non-consumers was simpler than that for the consumers, where more attributes, consequences and values were elicited. This might indicate the more complex, and possibly confused, product knowledge of the consumers, while obviously the non-consumers expressed simpler and more clear-cut cognition about why they disliked the product. Generally, all of the consumers associated sheep and goat meat with health at different levels of abstraction, and with enjoyment of life. In contrast, non-consumers felt disgusted and did not associate any health benefit to this kind of meat.

The consumer HVM suggests that the high-quality and unique taste of this meat are the two most important attributes. Sheep and goat meat consumers mostly liked the taste and the textural attributes (“Unique taste”, “Tenderness”), as has been reported in previous studies [9,20,22,36]. It is of note that these important attributes can be associated with a “sensory” dimension, which appears to be a distinctive characteristic of both sheep and goat meat. This dimension was also noted by de Andrade et al. [5], where consumers associated lamb meat with several positive hedonic attitudes, such as “tenderness”, “juiciness”, “flavour”, “tasty”, “delicious” and “yummy”. According to Bernués et al. [22], consumers also like cooking or eating sheep and goat meat to satisfy their guests and family. For this reason, consumers appear to perceive this meat as appropriate for special occasions, such as Sunday lunch with family, or holidays. This habit was also reported by de Andrade et al. [5], where the consumers associated lamb meat with special occasions (e.g., barbecues), in contrast to daily consumption.

The results of the present study show that, as for other types of meat, the taste of lamb meat is strongly related to a perception of high quality. Lind [26] investigated the purchase motivations for a specific kind of pork and reported that the quality perception was related to good taste, which led to the hedonistic value of enjoyment. However, quality is also associated with freshness of the meat [31,32]. Eating high-quality and fresh products is important for consumers, and freshness is usually seen as an important safety indicator [31]. The origin of the meat was another important quality cue. Consumers want to know the “region the meat comes from”, and they will preferably purchase and eat “domestic” meat [6,26]. 

Information about the origin, animal feeding, production systems and rearing conditions is another key factor [10,20]. As with Napolitano et al. [63,64], information of a more sustainable production system influences consumer acceptability of lamb meat, and thus should be included in marketing strategies.

Health is the other important value. When choosing sheep and goat meat, consumers satisfy their basic needs. Inclusion of sheep and goat meat in their diet was perceived as a healthy choice for a healthy life. Sheep and goat meat provide positive health benefits and have good nutritional value, which was mainly related to the perception of the animals being fed more naturally than some others, and (possibly) with a lower fat content compared to other meats [65]. Consumers, in general, associate free-range grazing and “natural” feed to greater quality and better health [20]. It is interesting to note that the sheep and goat meat was characterized as “Less fat” in the consumer HVM, while the attribute “Fatty” was mentioned by non-consumers as being among the barriers. On one hand, this may reflect the fact the usually younger ovine are consumed, as well as a specific acknowledgment that goat meat is leaner. On the other hand, the results indicate that non-consumers’ knowledge is quite limited.

Among the non-sensory attributes, price did not have a primary position in the mind of the consumer, and it can be considered as a minor quality indicator for meat [31]. Only non-consumers considered that price was an important barrier to the purchase and consumption [5,22].

For non-consumers, the strong taste and the smell represent the primary barriers to the consumption of sheep and goat meat. They felt a sense of disgust and disappointment or experienced a “gamey” taste and odour. Like for de Andrade et al. [5], those who had never eaten this kind of meat associated it with negative hedonic feelings, such as “pity” and “sorrow”. Additionally, Rousset-Akrim et al. [66] reported that some chemical compounds in this type of meat might produce or intensify the “gamey” odour, which is usually not well appreciated by non-consumers.

As already mentioned, the fat content was a clear barrier for non-consumers, contrary to consumers. Fat is directly connected to health, which was a central value, and was taken as a good reason to avoid the consumption of sheep and goat meat [67].

## 5. Conclusions

This international study contributes to the definition of the relevant motivations of both consumers and non-consumers towards sheep and goat meat. The findings indicate different levels of abstraction in the minds of the participants, with the focus on the motivations and barriers through the application of laddering and MEC approaches [38]. There are no other MEC studies in the literature at present in terms of consumer preferences for sheep and goat meat in European countries.

The results here highlight the importance of two main themes: health and the pleasure of eating. Consumers perceived sheep and goat meat as “tasty”, “natural” and “healthy” because of its lower environmental impact and fat content compared to other meats. In contrast, non-consumers mainly stressed the same chains, but considered that this meat is not healthy and not satisfying. Even though the results of the present study cannot be generalized, they do constitute the basis for future investigations. Further studies are needed to confirm our findings and to explore the antecedents of these attitudes in larger samples and on special populations looking for special attributes (e.g., Halal certification among the growing Muslim population).

However, from these findings, and as suggested by participants during the discussions, some strategies can already be suggested for increasing the consumption of this meat. First, communication of the benefits related to the healthiness of this meat and provision of more detailed information on the food labels in terms of the processing characteristics would be of help. Additionally, increasing the visibility of this meat (e.g., TV cooking programmes, magazine articles) and communication of the ease of its cooking are possible strategies for increasing the appeal of sheep and goat meat products in order to improve its consumption. Consumers were also interested in having information about the origin of the meat, and the production methods, traceability and possible positive effects on human health.

The qualitative nature of the focus group research techniques does not allow statistical generalization, that is, generalizing about specific population parameters. Focus group respondents were not selected to be representative of the population of respondents, but to maximize differences in responses, to collect wider points of view and to allow in-depth analysis of such differences. Additionally, each focus group may be considered as a single observation or case [68], while laddering questionnaires—albeit collected individually—provide a meaningful representation if summarized in terms of the most frequent and shared concepts and relative linkages. Nevertheless, focus groups make it possible to obtain a rich set of data on perceptions, thoughts, and opinions, while laddering makes it possible to study the network of concepts associated with the motivations and barriers for a specific consumer behaviour.

## Figures and Tables

**Figure 1 animals-10-01105-f001:**
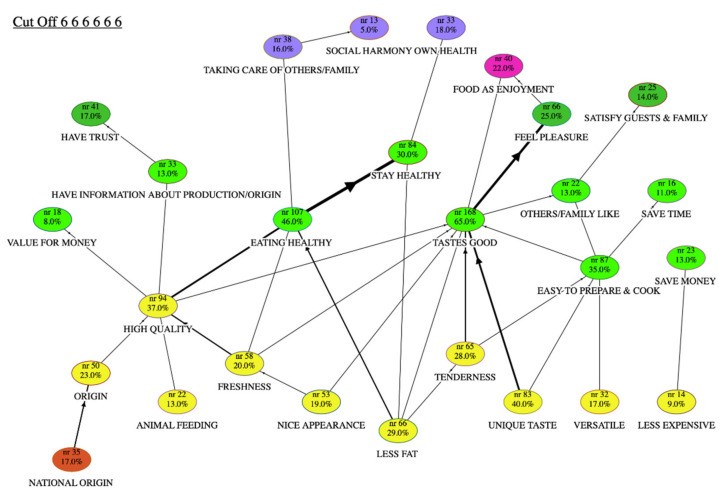
Hierarchical value map of sheep and goat meat consumers. For each node, the name and the number of answers are given in the content code, and the percentage of participants who named it.

**Figure 2 animals-10-01105-f002:**
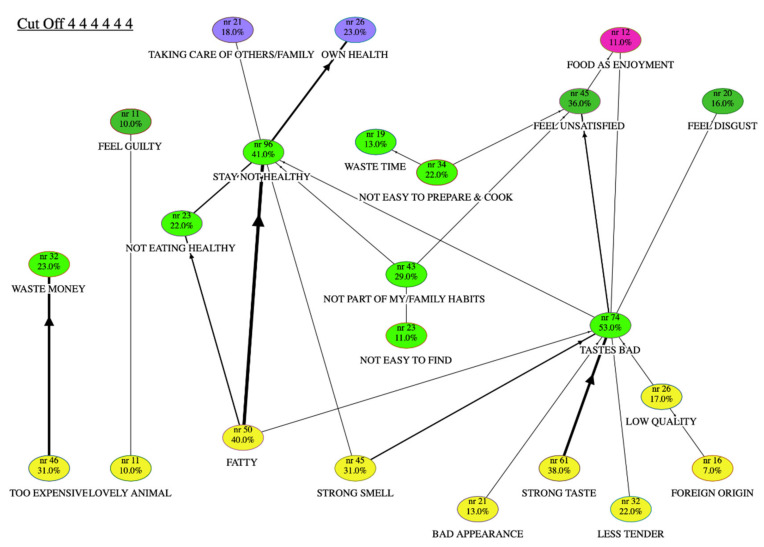
Hierarchical value map of non-consumers of sheep and goat meat.

**Table 1 animals-10-01105-t001:** Compositions of the focus groups by country.

Attribute	Detail	ES	FI	FR	GR	IT	TR *	UK	Total
Sex	Women	13	7	17	9	13	14	10	83
	Men	7	4	7	10	6	16	7	57
Age (years)	25–45	17	5	2	14	11	16	8	73
	46–65	3	6	22	5	8	14	9	67
Working	Yes	11	8	11	13	13	22	15	93
	No	9	3	13	6	6	8	2	47
Red meat consumption	Regular	12	1	13	8	9	18	10	71
	Occasional	8	10	11	11	10	12	7	69
Total		20	11	24	19	19	30	17	140

* 3 focus groups.

**Table 2 animals-10-01105-t002:** Description of the hierarchical value map content.

Sheep/Goat Maps	Cut-Off Level	Subjects	Ladders	Answers	Codes
Consumer motivation	6	139	463	1843	69
Non-consumer barriers	4	87	258	972	55

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
