# Peer review of "Motivations and Barriers for Sheep and Goat Meat Consumption in Europe: A Means–End Chain Study"

_animals, 2020, doi:10.3390/ani10061105_

Round 1

Reviewer 1 Report

The international study involving seven European countries (Italy, UK, Finland, France, Spain, Greece, Turkey) was conducted to explore the relevant reasons that affect both consumer and non-consumer perceptions towards sheep and goat meat. In total, there were 140 participants involved in the study.

According to the reviewer, the number of participants is too small, especially considering international research. It is impossible to draw conclusions from such a small sample.
Therefore, the research layout is incorrect.
According to the reviewer, in the Introduction section, there is no characterization of the properties of the meat in question, in terms of the content of, among others, individual nutrients. Especially since in the Conclusion section (line 491-493) the authors state that sheep and goat meat is " healthy" because of its lower content of fat compared to other meats.

Author Response

We have already tackled these comments in our first response.

We report here the relevant answers:

This was NOT a survey, but a qualitative research by use of group in-depth interviews (focus groups) and laddering questionnaires. To clarify, we have added the following paragraphs (lines 136-145):

Focus group research, as case study and other qualitative research methods, is conducted to gain a more complete understanding of a particular topic [1], such as consumer motivation, barriers and perceptions on sheep and goat meat in Europe. Laddering refers to “an in-depth, one-on-one interviewing technique used to develop an understanding of how consumers translate the attributes of products into meaningful associations with respect to self, following Means-End Theory”[2].The goal of both methods is not to achieve standard statistical generalizations but to go “in-depth” and achieve conceptual or analytic generalization, that is (a) corroborating, modifying, rejecting or otherwise advancing theoretical concepts or (b) providing new concepts, ideas, theories, hypotheses [3]. This form of generalisation is driven by semantics rather than statistics [4].

The qualitative nature of the focus group research techniques does not allow statistical generalization, that is to generalize about specific population parameters [5]. Focus groups respondents are not selected to be representative of the population of respondents, but to maximize differences in responses, to collect wider points of view and to allow in-depth analysis of such differences.  We have been very careful about not making such generalizations. However, not all generalization from focus groups is inappropriate. As experiments, focus group research is useful to answer “How” “Why” research questions [3].  Focus groups are great to elicit “discourses” and “viewpoints” on a topic. Given the length and depth of the discussion held in the focus groups and the careful design and use of probing questions, we can be quite sure that only very minor consumptions motivations and barriers might have not showed up. Still, we acknowledge that we refer to most relevant motivations and barriers (line 442ss) and have acknowledged the limitations of the method (see lines 521ss)

Reviewer 2 Report

The manuscript dealing with a study on consumer preference and barriers for sheep and lamb consumption. It is based on a qualitative analysis carried out in seven European countries where are investigated the relevant reasons that affect both consumer and non-consumer perceptions towards sheep and goat meat.

The study is interesting and well-structured, other than easy to read thanks also to the fluency of the text. Few studies dealing with this topic and thanks to its originality surely it enriches the literature on consumers' preferences on meat consumption. I strongly encourage its publication on Animals.

However, there are a few points that need to be addressed to improve the important contribution that this manuscript offers to the international literature.

- Material and Method section - line 130: here you mention to regular and occasional consumers of sheep and goat meat products, but it is not clear how you asked about the frequency of consumption. I mean, are regular consumers those eating these meat products every day or at least 3 times per week? And similarly, who are occasional consumers? Please, explain what do you mean for regular and occasional consumers. You can also report this frequency into brackets.

- Conclusion section: maybe some suggestions about further researches could be reported.
(For example, but it is only my opinion, that the high frequency of consumption of sheep and lamb meat products in some countries is also strictly linked to some religious communities -which are growing in many European countries-. In this regard, some readings would suggest that the Muslim community represents an important consumer base for lamb. Therefore, further research could go in this direction, taking into account also the halal certification.).

Author Response

We would like to thank the reviewer for fruitful suggestions and comments.

line 130ss: we have now added this sentence:

Consumer classification was based on self-reported frequency of consumption: regular consumers were those who reported consuming sheep/goat meat at least one time per month, occasional consumer less than so but not never or very seldom.

Conclusions: we have now added the following sentence on future research as suggested:

Further studies are needed to confirm our findings and to explore the antecedents of these attitudes in larger samples and on special populations looking for special attributes (e.g. Halal certification among the growing Muslim population).

Reviewer 3 Report

A very interesting paper, aimed to fill the gap of knowledge about small ruminant meat consumption. After a full and comprehensive literature review, Authors analysed consumer perceptions of sheep and goat meat, providing a qualitative description of the motivations and barriers in goats and sheep meet consumption.

According to my opinion this paper is worthy of consideration and to be published for several reasons: it allows to gain information on consumer’s behaviour, meanwhile consumption paths for goats and sheep meats are still relatively unexplored; the results are of particular interest because they rise from direct inquires developed simultaneously in seven European Countries; the Authors adopted a combination of two qualitative approaches (focus groups and means-end chain (MEC)). Both methods, and particularly their combined implementation are, in my opinion, perfectly suitable to provide new insight about consumer acceptance of sheep and goat meat. Focus groups results clearly show that consumers’ motivations, barriers and opinion about meat consumption were carefully considered, elicited and taken into account in results reporting. Finally, the MEC methodology allowed to map consumer’s behaviours defining the relations among product attributes, benefits and consumer’s value and suggesting drivers and hindrances to goat and sheep meats consumption.

The authors provided a full reply to the previous comments and amended the text accordingly where necessary.

In the manuscript there are two minor misprint:

Line 259 “rekawted” instead of related

Lines 484-486 The sentence “On one and……..that non consumers”that is not reported in the former version seems to be a misprint to be deleted

Author Response

We thank the reviewer for his/her comments.

We have corrected the typo and completed the sentence, that now reads as follows:

On one hand, this may reflect the fact the usually younger ovine are consumed, as well as a specific acknowledgment that goat meat is leaner. On the other hand, the results indicate that non-consumers’ knowledge is quite limited.